# Degraded RNA from Human Anterior Cruciate Ligaments Yields Valid Gene Expression Profiles

**DOI:** 10.3390/ijms24031895

**Published:** 2023-01-18

**Authors:** Megan N. Ashton, Asha E. Worsham, Matthew D. Strawn, Geoffrey D. Fisher, Cody J. Perry, Matthew P. Ferguson, Mimi Zumwalt, George W. Brindley, Javad Hashemi, Hossein Mansouri, James R. Slauterbeck, Daniel M. Hardy

**Affiliations:** 1Department of Cell Biology & Biochemistry, Texas Tech University Health Sciences Center, Lubbock, TX 79430, USA; 2Department of Anesthesiology, Virginia Commonwealth University, Richmond, VA 23298, USA; 3Department of Orthopaedic Surgery & Rehabilitation, Texas Tech University Health Sciences Center, Lubbock, TX 79430, USA; 4Department of Ocean & Mechanical Engineering, Florida Atlantic University, Boca Raton, FL 33431, USA; 5Department of Mathematics & Statistics, Texas Tech University, Lubbock, TX 79409, USA; 6Department of Orthopaedic Surgery, University of South Alabama, Mobile, AL 36604, USA

**Keywords:** ligament, tendon, biomechanics, gene expression, RNA, knee, tissue remodeling

## Abstract

Correlating gene expression patterns with biomechanical properties of connective tissues provides insights into the molecular processes underlying the tissue growth and repair. Cadaveric specimens such as human knees are widely considered suitable for biomechanical studies, but their usefulness for gene expression experiments is potentially limited by the unavoidable, nuclease-mediated degradation of RNA. Here, we tested whether valid gene expression profiles can be obtained using degraded RNA from human anterior cruciate ligaments (ACLs). Human ACL RNA (N = 6) degraded in vitro by limited ribonuclease digestion resemble highly degraded RNA isolated from cadaveric tissue. PCR threshold cycle (C_t_) values for 90 transcripts (84 extracellular matrix, 6 housekeeping) in degraded RNAs variably ranged higher than values obtained from their corresponding non-degraded RNAs, reflecting both the expected loss of target templates in the degraded preparations as well as differences in the extent of degradation. Relative C_t_ values obtained for mRNAs in degraded preparations strongly correlated with the corresponding levels in non-degraded RNA, both for each ACL as well as for the pooled results from all six ACLs. Nuclease-mediated degradation produced similar, strongly correlated losses of housekeeping and non-housekeeping gene mRNAs. RNA degraded in situ yielded comparable results, confirming that in vitro digestion effectively modeled degradation by endogenous ribonucleases in frozen and thawed ACL. We conclude that, contrary to conventional wisdom, PCR-based expression analyses can yield valid mRNA profiles even from RNA preparations that are more than 90% degraded, such as those obtained from connective tissues subjected to biomechanical studies. Furthermore, legitimate quantitative comparisons between variably degraded tissues can be made by normalizing data to appropriate housekeeping transcripts.

## 1. Introduction

Profiles of expressed mRNAs reflect the physiological states of cells and tissues [1,2]. Consequently, comparative expression profiling can identify genes and metabolic pathways associated with the onset, progression, and treatability of disease [3,4]. Approaches to mRNA profiling range from global analyses, such as RNA-Seq or microarray screening of all expressed genes, to a more focused, PCR-based profiling of gene subsets. Regardless of the approach, expression profiling methods universally call on the use of only high-quality, intact RNA to ensure that results accurately reflect the pattern of mRNAs present in living cells or tissues [5,6]. Appropriate methods for tissue acquisition and handling and for RNA extraction and purification can yield intact, high-quality RNA, even from difficult tissues such as the pancreas and ligaments [7,8]. However, RNA degradation prior to isolation is sometimes unavoidable in precious specimens such as human cadaveric tissues, which cannot always be handled and stored under ideal conditions, thus potentially limiting the usefulness of RNA for these tissues for expression analyses.

RNA profiling studies using fresh or fresh–frozen tissues have characterized the molecular responses of connective tissues to injury in non-human animals [8,9], and identified sex differences in the expression of tissue remodeling genes, including collagens, matrix metalloproteases (MMPs), and tissue inhibitors of MMPs (TIMPs) that may underlie the susceptibility of female athletes to anterior cruciate ligament (ACL) injury [10,11,12,13,14]. However, RNA integrity often cannot be completely preserved in biomechanical studies of human cadaveric tissues, which confounds studies seeking, for example, to correlate gene expression with the human ACL ultrastructure and biomechanical properties [15,16,17,18] or to identify expression patterns associated with the degeneration and biomechanical properties of human intervertebral discs [19,20]. Some studies have shown that meaningful expression data can be obtained using degraded RNA from sources, such as breast cancer cell lines [21,22], paraffin-embedded tumor biopsies [23], shed bladder tumor cells [24], and rectal tumors [25], but no studies have assessed the validity of profiles generated using degraded RNA from human connective tissues commonly used for biomechanical studies. Here, we tested the hypothesis that gene expression profiles generated using degraded RNA from human ACLs, accurately reflect those generated using intact RNA.

## 2. Results

Limited RNase digestion of ACL RNAs yielded degraded preparations with electrophoretic properties similar to highly degraded RNA isolated from human cadaveric intervertebral disc (Figure 1). Notwithstanding identical digestion conditions, the six degraded ACL RNA preparations exhibited subtle electrophoretic differences, with RNAs from ACL 98 and 104 appearing less degraded than the other four RNAs (Figure 1).

The expression profiling by real-time RT-PCR quantified mRNAs of 89 extracellular matrix components, including 13 MMPs, 12 collagens, and 3 TIMPs (Appendix A). Using hierarchical models, a regression analysis of C_t_ pairs (threshold cycle values, degraded vs. intact) for all 89 genes in RNA from the six ACLs revealed a strong linear relationship (Pearson correlation coefficient r = +0.80) provided by the following equation:*C*_t__*degraded* = 5.769 + 0.906 × *C*_t__*intact*


The line intercept (95% confidence interval 3.254–8.284 cycles) reflected an increase in the number of PCR cycles required to reach the threshold signal as a consequence of degradation-induced decreases in the amounts of amplifiable mRNAs for the 89 targets. The line slope near, but slightly below, 1.0 (95% confidence interval: 0.868–0.943) revealed that degradation produced a generally uniform increase in C_t_ irrespective of an individual RNA’s abundance, with a possible trend toward a proportionally greater loss of higher abundant mRNAs.

A hierarchical analysis of individual regression lines for the six RNA preparations (Figure 2) had strong linear relationships (range of Pearson coefficients: 0.86–0.98; Table 1), with non-random slopes of the six lines each fixed at 0.906 for the overall population.

In contrast to the common slopes of the regression plots, randomly distributed intercepts ranged from 2.93 to 7.48, possibly reflecting a stochastic variation in the extent of RNA degradation between preparations. Mean C_t_ increase ranged from 0.75 cycles for the 86 targets detected in degraded ACL 104 RNA to 5.2 cycles for the 80 targets detected in degraded ACL 105 RNA, corresponding to 40% and 97% average losses of amplifiable target, respectively, as calculated using the following formula:Fraction remaining = *f* = 2exp(−ΔC_t_)  % loss = (1 − *f*) × 100(1)

The randomly distributed y-intercept (b) values inversely correlated with the number of quantified targets (N) and with the magnitude of the Pearson coefficient (r), indicating that, as the average C_t_ increased because of RNA degradation, more targets were lost and the scatter of the data increased. Nonetheless, even at the highest levels of degradation (ACLs 99, 105, 108, and 125), profiles from individual degraded RNAs strongly correlated with those from the corresponding intact RNAs (r ≥ +0.86). Likewise, the coefficients of determination (R^2^) ranging from 0.75 to 0.96 indicated that 75–96% of the variation in target signals from degraded RNAs reflected the biologically relevant distribution of target signals from corresponding intact RNA.

To determine if the RNase digestion of RNA in vitro accurately modeled the degradation of RNA in poorly stored tissues, we thawed ACL tissue to permit the action of endogenous RNases in situ prior to RNA isolation and compared the gene expression profile from the degraded RNA to that obtained with intact RNA. Thawing 5–20 min before initiating extraction induced overt degradation, with progressive decreases in the amounts of intact rRNAs, decreases in the ratio of 28S to 18S rRNA, and corresponding increases in the amounts of smaller digestion products in comparison to RNA from unthawed ACL (Figure 3A,B). Nevertheless, as observed for in vitro digested RNAs, the gene expression profile obtained using RNA from thawed ACL strongly correlated with the profile from unthawed ACL (R^2^ = 0.92). Furthermore, in situ digestion produced a slope of the degraded vs. intact regression line (0.915, Figure 3C) very close to slope of 0.906 from the in vitro digestion experiments.

To examine the variability in the quality of RNA in cadaveric tissues, we isolated total RNA from two human testes frozen within 6 h of acquisition, and conducted fragment analysis in parallel with formaldehyde agarose gel electrophoresis to enable comparisons to degraded RNAs from ACL (Figure 1, Figure 3 and Figure 4). Both tissues yielded RNA preparations with quality/integrity superior to that of in vitro-and in situ-degraded ACL RNAs used for expression analyses (Figure 2 and Figure 3).

For each of our in vitro-degraded RNA preparations, mean increases in C_t_ for housekeeping and non-housekeeping genes did not differ (*t*-tests; 0.25 < *p* < 0.40). Furthermore, a regression analysis revealed a strong correlation (r > 0.999; Figure 5) between C_t_ increases in housekeeping and non-housekeeping genes among the six degraded RNA preparations, spanning a wide range of mean C_t_ increases corresponding to levels of RNA degradation from approximately 50% to more than 95%. Furthermore, in situ digestion produced very similar losses of housekeeping and non-housekeeping gene mRNAs (mean C_t_ increases of 2.2 and 2.3, respectively, reflecting more than a 75% loss of amplifiable targets in the tissue thawed for 20 min), comparable (red data point in Figure 5) to the relative losses observed in the in vitro digestion experiments.

## 3. Discussion

Here, we found that the degradation of human ACL RNA decreased the amount of individual mRNAs amplified in RT-PCR-based expression analyses but did not dramatically alter the overall gene expression profile in comparison to intact RNA. We also found no significant difference between the loss of housekeeping and loss of non-housekeeping mRNAs within degraded RNA preparations. Therefore, by normalizing to levels of housekeeping mRNAs, legitimate comparisons can be made between profiles of mRNAs with varying levels of degradation. Indeed, up to a 97% mean loss of amplifiable targets still yielded profiles that strongly correlated with those obtained from intact RNA, albeit with a loss of signals from some low-abundance targets. Further digestion of RNA is required to define the point at which the excessive degradation of ACL RNA yields invalid expression profiles. Regardless, our results reveal that meaningful, PCR-based expression profiles can be generated from RNA that is much more highly degraded (RQN/RIN well below 4) than is generally thought to be acceptable for such studies. Our findings are consistent with those of prior studies examining smaller gene sets in degraded RNAs from various tumor cells and tissues [21,22,23,24,25,26].

ACL RNA digested in vitro with RNase ONE™ could differ from RNA degraded by endogenous cellular ribonucleases. RNase ONE™ is a broad specificity enzyme from *E. coli* that cleaves between any two ribonucleotides, whereas human cellular ribonucleases exhibit more restricted nucleobase specificities [25]. Nevertheless, differences in enzyme action are unlikely to affect our findings for two reasons. Firstly, the promiscuous activity of RNase ONE™ theoretically approximates the combined specificities of the multiple RNases [26] in human cells that together mediate in situ RNA degradation in a poorly handled or stored tissue. Secondly, the hydrolysis of an mRNA anywhere between the upstream PCR primer sequence and the downstream priming site for reverse transcriptase would render it incapable of serving as a template for RT-PCR. Consequently, any endogenous RNase, including one that preferentially hydrolyzed at certain ribonucleotides (such as the CpX specificity of pancreatic RNase A [27]), would be expected to sufficiently digest target mRNAs, causing the loss of RT-PCR signal that we achieved by digestion with RNase ONE™. Finally, expression analysis of RNA degraded in situ in frozen-and-thawed ACL, which mimics the likely conditions of a poorly stored cadaveric tissue, yielded nearly identical results, thus confirming the conceptual soundness of the in vitro digestion approach in this study.

Our results do not rule out the possibility that certain individual RNAs might be highly sensitive to hydrolysis by specific RNases; however, the use of RNA that is presumed to be completely intact also does not necessarily preclude that possibility. Notably, using a combination of oligo-dT and random hexamers to prime the RT step for cDNA synthesis, we maximized yields of first-strand template for downstream amplification by PCR. Thus, even RNA preparations entirely degraded to relatively small fragments could still generate expression signals so long as individual mRNA fragments spanned their corresponding PCR-amplified target sequences and extended far enough downstream to provide a site for oligo-dT or random hexamer priming. Accordingly, the results of this study show that useful expression information for most genes can be obtained from a human connective tissue acquired in a way that does not assure the complete integrity of RNA (e.g., from biopsies or cadaveric specimens).

Our experiments did not examine RNA degradation in poorly stored, cadaveric ACL per se, but instead modeled degradation in surgically acquired specimens. This experimental approach assured that the RNA we deemed “intact” has the highest possible integrity/quality. In practice, ACLs from deceased donors could seldom, if ever, be preserved to block RNA degradation as quickly as ACLs excised in the OR from a replaced knee joint and then immediately frozen in N_2_(*l*). Additionally, whether a tissue itself is “alive” or “dead” is independent of whether the donor was alive at the time of acquisition, thus confounding possible distinctions between surgical and cadaveric specimens. To this end, RNAs isolated from cadaveric testes frozen within 6 h after donor death illustrated the degradation anticipated in such tissues, with measured RIN/RQN values generally considered marginal (RQN = 7.0) and unacceptable (RQN = 4.3) for downstream expression studies. Notwithstanding those norms, our in vitro digestion experiments showed that useful expression data may be obtained from much more highly degraded RNA preparations (RIN/RQN approaching 1).

Although RNA in poorly stored or handled tissues will ultimately degrade to the point where the valid expression data cannot be obtained, we conclude it is possible to generate meaningful results from degraded RNA so long as a sufficient mRNA template remains for PCR amplification. This finding is particularly relevant to research correlating gene expression profiles with the biomechanical properties of cadaveric tissues, including studies on sex differences in molecular and biomechanical properties of the human ACL [10,11,12,13,14,15,16,17,18], and on age- and pathology-associated differences in the molecular and biomechanical properties of intervertebral discs [19,20].

## 4. Materials and Methods

Tissue acquisition. With patient consent and per IRB-approved protocol (TTUHSC IRB# L06-154), we recovered human ACLs normally discarded during total knee arthroplasty (8 male, 5 female; aged 29–73 years; mean: 62.3 years), and immediately froze [N_2_
*(l)*] the tissues for storage at −80 °C. Additionally, per IRB# L06-154, clinical staff removed protected health information (Pt name, date of procedure, details of diagnosis, etc.) to de-identify specimens prior to transfer to the research lab. For a cadaveric comparison tissue, we excised the L4-5 disc from the frozen (−20 °C) spine of a willed body (non-embalmed female, age 40; obtained from the UT Southwestern Willed Body Program, https://www.utsouthwestern.edu/research/programs/willed-body/, accessed on 18 December 2022 with donor consent for use in research). As is commonplace for donated bodies, no information was available regarding the time between donor death and freezing of the spine, or the specific conditions of its storage prior to receipt in the lab. Finally, for additional, RNA-rich comparison tissue, we used cadaveric human testes (N = 6), acquired with consent for use in research by the National Disease Research Interchange (https://ndrireseource.org, accessed on 18 December 2022), and frozen within 6 h of donor death.

RNA isolation. We isolated RNA from frozen [N_2_ (*l*)], thoroughly pulverized tissues (whole ACL, outer anterior segment of the intervertebral disc’s annulus fibrosus, or cadaveric testes using either the RNeasy Mini Kit (100–350 mg of tissue; QIAGEN, Valencia, CA, USA) or the guanidinium thiocyanate/acidic phenol extraction [7,8] method (1 g tissue); assessed RNA purity and concentration by spectrophotometry (NanoDrop™, Thermo Fisher Scientific, Wilmington, DE, USA); and assessed RNA quality/integrity by formaldehyde-agarose gel electrophoresis [28]: Agilent 2100 Bioanalyzer (Santa Clara, CA, USA) or Agilent 5200 Fragment Analyzer and ProSize Data Analysis software.

RNA degradation. For in vitro degradation of ACL RNA, we digested paired aliquots (4 µg, isolated by RNeasy kit) either with RNase ONE™ (0.005 units, 37 °C, 1 h; Promega, Madison, WI, USA) to produce “partially degraded” RNA, or with no added enzyme to produce a mock-digested, “intact” control RNA, and assessed extent of degradation by formaldehyde/agarose gel electrophoresis (1 µg RNA/lane) [28].

For the in situ degradation of ACL RNA, we incubated 1 g of thawed, pulverized ACL for 5, 10, 15, and 20 min at 23 °C to permit the autolytic degradation of RNA by endogenous RNAses, isolated RNA by the guanidinium thiocyanate/acidic phenol extraction method [7,8], and then assessed RNA quality with an Agilent Technologies 2100 Bioanalyzer. A matching 1 g portion of unthawed, pulverized ACL served as a non-degraded control.

Quantitative PCR. We used first-strand complementary DNA (cDNA) synthesized from 500 ng of RNA with combined oligo-dT and random hexamer priming (RT2 First Strand Kit; SABiosciences, Frederick, MD, USA) as a template for real time PCR (Human Extracellular Matrix and Adhesion Molecules RT2 Profiler™ PCR Array kit; SABiosciences), with SYBR Green detection. We calculated threshold cycle (C_t_) values at threshold = 0.4 from amplification plots (ABI Prism 7000 Sequence Detection System; Applied Biosystems, Foster City, CA, USA), and omitted from further analysis any genes that the software identified as “undetermined” because they were not sufficiently amplified during the exponential phase. All five housekeeping genes in the array were successfully amplified in all profiles; therefore, downstream correlation analyses (see below) included C_t_ values from the full complement of these genes.

Data analysis. To determine the effect of RNA degradation on expression profiles, we correlated C_t_ values obtained from degraded RNAs and non-degraded RNAs using hierarchical regression models. We first established a linear relationship by analyzing all data from the six ACL RNA preparations, then produced individual regression equations for ACL RNAs by treating them as random samples from the population regression, and examined the assumption of randomness of the regression coefficients by fitting hierarchical linear regressions with random slopes and random intercepts. To analyze the variation in degradation, we averaged the differences in C_t_ values between intact and degraded RNAs for each gene in each of the six ACLs, then plotted the averages for housekeeping and non-housekeeping genes against each other to test for correlations. Finally, we performed two-tailed *t*-tests to identify differences in extent of degradation between the housekeeping and non-housekeeping groups.

## Figures and Tables

**Figure 1 ijms-24-01895-f001:**
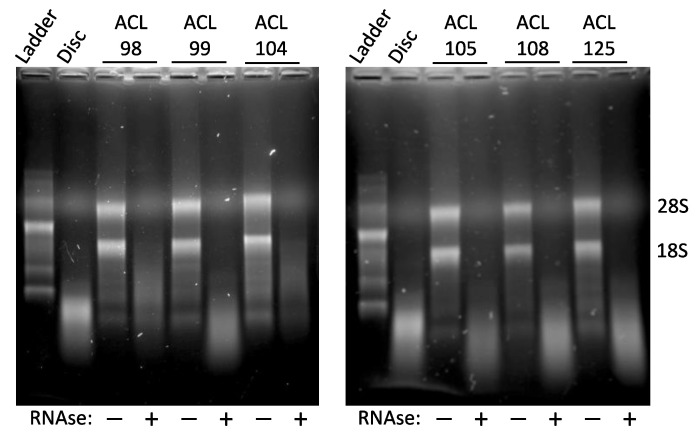
Electrophoretic comparison of intact and degraded RNA preparations. Formaldehyde agarose gel profiles of paired intact (RNAse −) and in vitro-degraded (RNAse +) RNAs from six human ACLs are shown, designated 105, 108, 125, 98, 99, and 104, recovered with consent during total knee arthroplasty (TKA). Note the extensive degradation produced by RNAse digestion (+), evident in the loss of the 28S and 18S ribosomal RNA bands and increased abundance of smaller fragments in comparison to mock-digested RNA (−), as well as the electrophoretic similarity to RNA isolated from cadaveric intervertebral discs. Note the lower abundance of smallest RNA fragments from digestion of ACLs 98 and 104, reflecting the relatively lower extensive degradation of those RNAs.

**Figure 2 ijms-24-01895-f002:**
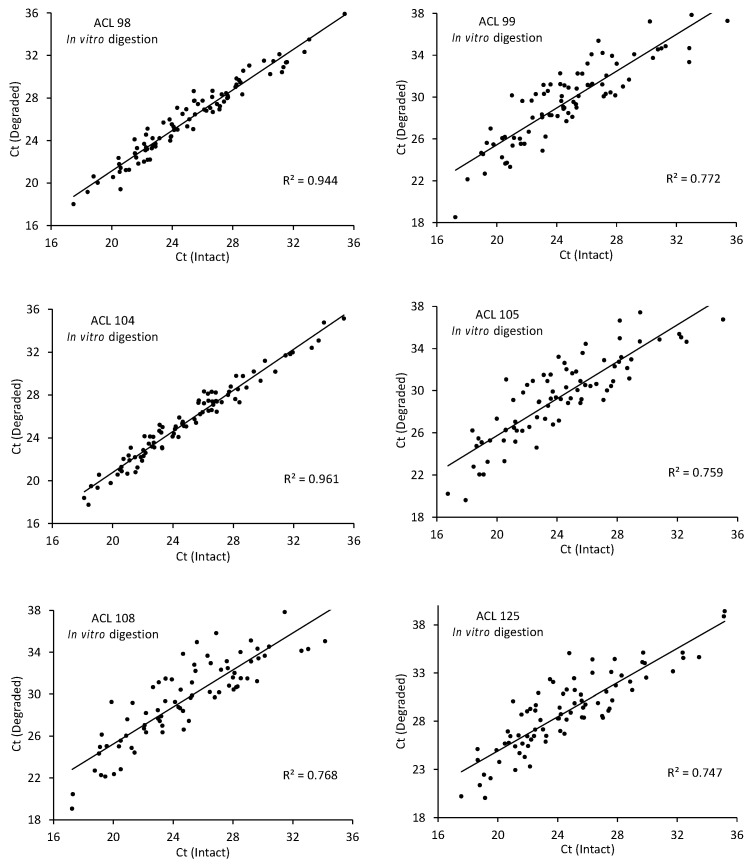
Linear regression analysis of C_t_ values for 80+ targets quantified in intact vs. in vitro-degraded RNA from the six ACLs in Figure 1. See Table 1 for a summary of regression parameters. Note the lower scatter and correspondingly higher correlations observed for the two relatively less degraded RNA preparations (ACLs 98 and 104).

**Figure 3 ijms-24-01895-f003:**
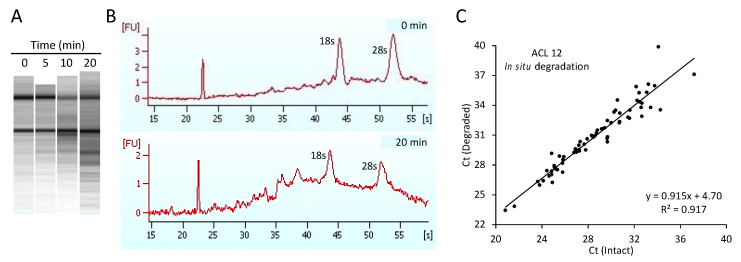
Comparison of expression profiles from intact and in situ-degraded RNA. (**A**): Bioanalyzer gel images illustrating the time course of in situ RNA degradation in ACLs thawed and held at 23 °C for 5, 10, or 20 min prior to RNA extraction. Note the progressive loss of 28S rRNA and accumulation of numerous smaller, discrete digestion products in comparison to the RNA isolated coincident with thawing (0 min). (**B**): Bioanalyzer traces of the RNA preparations isolated 0 min (upper) or 20 min (lower) post-thaw. The peaks at 44 and 52 min are the 18S and 28S ribosomal RNAs, respectively. For the RNA-isolated 20 min post-thaw, note the increased baseline and smaller rRNA peaks, reflecting the autolytic degradation of RNAs to smaller products. (**C**): linear regression plot of C_t_ values from the intact vs. in situ-degraded RNAs. Note the strong correlation (R^2^ = 0.917) between the profiles and the very similar slope of the regression line (0.915) to that observed in in vitro digestion experiments (Table 1).

**Figure 4 ijms-24-01895-f004:**
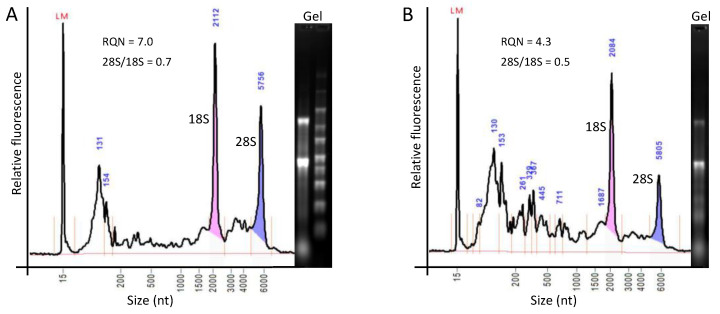
Agarose gel and fragment analyzer assessment of RNA integrity/quality in human cadaveric tissues. (**A**,**B**) show results from two human testes frozen within 6 h of donor death. RQN = RNA Quality Number (fragment analyzer equivalent of RNA Integrity Number, RIN). LM = Lower Marker. Shading of 18S and 28S rRNA peaks depicts areas under curve used for calculating RQN and 28S/18S ratio. Note the distinct rRNA bands and differences in RQN and 28S/18S (which approach 10 and 2 in fully intact RNA, respectively) in both the agarose gel images and the fragment analyzer traces between the two tissues, as well as the superior quality of the RNAs in comparison to the in vitro-digested ACL RNAs used for expression analyses (Figure 1 and Figure 2).

**Figure 5 ijms-24-01895-f005:**
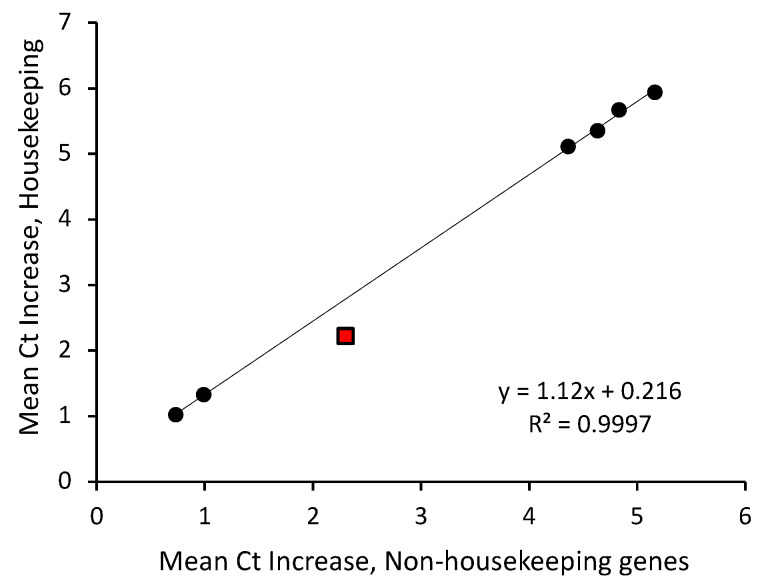
Effects of RNA degradation on mean C_t_ values for housekeeping and non-housekeeping genes among the 89 targets evaluated. Regression analysis revealed a linear relationship spanning a range of C_t_ increases from approximately one cycle in the least degraded preparations (corresponding to ~50% mean loss of target mRNAs) to approximately six cycles in the most degraded preparations (corresponding to >95% mean loss of target). Black circles: in vitro RNA digestions. Red square: in situ RNA degradation in frozen and thawed ACL.

**Table 1 ijms-24-01895-t001:** Correlation of mRNA expression profiles from partially degraded ACL RNAs with profiles from their corresponding intact RNAs. Differences in N (number of genes compared) reflect differences in the number of C_t_ values returned for each ACL, with N = 89 representing no “undetermined” targets for an individual ACL owing to the degradation of RNA target template to a level below that required for PCR amplification. Note that all six ACL RNAs yielded lines with a slope of 0.906, the same as for the combined data, but with randomly distributed y-intercept values (b). Note the higher coefficients of determination, reaching a very strong positive correlation of 0.96, for all individual lines in comparison to the combined data.

ACL	Targets Compared (N)	Slope (m)	y-Intercept (b)	Pearson Coefficient (r)	Coefficient of Determination (R^2^)
Combined	497	0.906	5.77	+0.80	0.64
98	87	0.906	3.28	+0.97	0.94
99	83	0.906	7.19	+0.88	0.77
104	86	0.906	2.93	+0.98	0.96
105	80	0.906	7.48	+0.87	0.76
108	78	0.906	7.01	+0.88	0.77
125	83	0.906	6.72	+0.86	0.75

## Data Availability

Raw data are available by written request to the corresponding author.

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
