# Peer review of "Degraded RNA from Human Anterior Cruciate Ligaments Yields Valid Gene Expression Profiles"

_ijms, 2023, doi:10.3390/ijms24031895_

Round 1
Reviewer 1 Report
It is not clear if the authors would like to target the problem of partial RNA degradation due to storage of cadaveric samples or after biomechanical test of the samples. The project title is on ACL. However, no cadaveric ACL was used for comparison. Therefore, I am not sure if the authors would like to examine the impact of RNA degradation after biomechanical test or storage of cadaveric samples.
To determine the RNA quality, an RNA integrity number (RIN) or DV200 should be calculated.
While I agree that gene sequencing and qPCR possibly can tolerated some partial RNA degradation, the specific conditions and the degree of RNA degradation that would still yield valid data were not examined and defined in this study. The threshold of RNA degradation that still produces valid data was not determined.
The lack of information of how the cadaveric disc sample was preserved and how long it has been preserved is a limitation of this study.
The use of RNase for partial RNA digestion is artificial. The authors should use cadaveric samples for this study.
The authors determined if RNase digestion of RNA in vitro accurately modeled the in-situ degradation of RNA in poorly stored tissues. Typical cadaveric samples should be used.
Although the authors showed that the mRNA loss was similar between housekeeping and non-housekeeping, the larger Ct values for genes in degraded samples would affect the reliability of the results.
Labels in the figures overlapped and cannot be seen clearly.
Author Response
We appreciate the reviewer's efforts in evaluating our manuscript, and believe our responses to the concerns raised have improved our study.
- We purposely made no distinction between RNA degradation from tissue storage or mechanical testing because both are relevant to our larger purpose of examining degraded RNA in ACL and other connective tissues. We believe our study addresses the question of these tissues' utility for gene expression studies regardless of the source of RNA degradation.
- We added a figure (new Fig. 4) illustrating RQN/RIN for RNAs isolated from cadaveric tissue for comparison to our in vitro- and in situ-degraded preparations (which are no longer available for direct analysis). These data show that our degraded ACL RNA preparations have RQN/RIN far below the generally accepted threshold for gene expression studies.
- We provided all information available for the cadaveric disc sample, and believe the relative lack of information vividly illustrates the problem our study addresses: that it's largely impossible to prevent RNA degradation in cadaveric tissues prior to receipt in the lab, but the tissues can still yield useful gene expression data.
- We believe our in vitro and in situ digestion/degradation studies using surgical specimens reliably models RNA degradation in cadaveric tissue, as explained in the second paragraph of the Discussion. It should be noted that the state of the tissue is most dependent on the time and conditions preceding its preservation by freezing or some other means, regardless of whether the tissue came from a living or dead donor. We also now explain that with cadaveric tissues it's impossible to assure we have the truly intact RNA needed for comparison.
- Yes, we noted that as extent of degradation increased, Ct values also increased as expected, and the scatter of the data increased. But the relative expression of the 80+ genes still correlated nicely between degraded and intact RNAs, especially for preparations that were about 50% degraded, which is much more highly degraded than samples normally considered necessary for gene expression profiling (RQN/RIN in the range of 6-7 or above).
- Figure labels have been revised as suggested.
Reviewer 2 Report
Thank you for presenting this interesting work on ACL RNA degradation. I have a few suggestions to strengthen the manuscript prior to publication listed below.
Lines 52-56: Please clarify the nature of the tissues in these studies (i.e. post-mortem, fresh frozen, etc) as this is relevant to your research study.
Lines 58-60: Citations to "unpublished observations" are unclear. If possible, please reference other published works, data presented in this paper, or conference abstracts.
Lines 67-68: Please note the treatment history (cadaveric, fresh frozen, etc) for the ACL RNAs and partially degraded RNA in this sentence, as the reader will not see this in the methods until much later in the manuscript.
Figure 1: The labeling on this image is difficult to interpret. Please replace the labels such that symbols are not complicating the text.
Figure 2: Plot titles and axis labels again have issues with symbols in the text.
Line 113: The equation should be isolated from the text in the paragraph, as was done for Equation 1 on line 82.
Figure 3: I believe the Y axis label should read "Mean" Ct increase.
Was any repeatability work done on samples from a single specimen in order to look at intra-subject variability?
Line 216-219: Was the tissue stored at -80 after dissection to match the ACL tissue? Or was it kept at -20?
Author Response
We greatly appreciate this reviewer's effort in evaluating our manuscript, and believe our responses to concerns have improved the study. Specifically, we have:
- clarified the nature of tissues as requested.
- added relevant citations.
- supplied the requested information in the Fig. 1 legend.
- revised Fig 1 labeling to improve presentation.
- expanded Fig 2 to include all regressions, and revised labels as requested.
- isolated the equations on their own lines as suggested.
- corrected the Y axis label as noted.
- not added data on intra-subject variability because the study focused on effects of RNA degradation. We appreciate this suggestion, and would certainly include such biological replicates if our purpose had been to make comparisons of specific genes between individuals.
- kept the language regarding the dissected disc as-was because the following paragraph stated that tissues were pulverized after freezing in liquid N2, which we hope makes clear how we processed the disc.
Round 2
Reviewer 1 Report
There is no doubt that some markers are more resistant to degradation and some useful information may still be obtained from the degraded RNA samples. While this study demonstrates this, there is no generalized rules of the acceptable quality of RNA or marker genes that would yield valid results. It has to be determined case by case.
The gene set that has been amplified should be provided in supplementary information.
Author Response
We greatly appreciate the reviewer's suggestion to include the gene set as supplemental information, and will do so (assuming the system provides the option for the upload). We have accordingly revised the manuscript to include text referring to the gene set table.
